# Association between sense of coherence and depression in patients with chronic pain: A systematic review and meta-analysis

Alejandra Aguilar-Latorre[1,2], Ángela Asensio-Martínez[1,2,3], Bárbara Oliván-Blázquez[1,2,3]*, Celia Álvarez-Bueno[2,4,5], Iván Cavero-Redondo[2,4,6], Christos Lionis[7], Emmanouil K. Symvoulakis[7], Rosa Magallón-Botaya[1,2,8]

**1** Institute for Health Research Aragón (IIS Aragón), Zaragoza, Spain, **2** Research Network on Chronicity, Primary Care and Health Promotion (RICAPPS, RD21/0016/0005), Carlos III Health Institute, Madrid, Spain, **3** Department of Psychology and Sociology, University of Zaragoza, Zaragoza, Spain, **4** Health and Social Research Center, Universidad de Castilla-La Mancha, Cuenca, Spain, **5** Universidad Politécnica y Artística del Paraguay, Asunción, Paraguay, **6** Facultad de Ciencias de la Salud, Universidad Autónoma de Chile, Talca, Chile, **7** Clinic of Social and Family Medicine (CSFM), School of Medicine, University of Crete, Heraklion, Crete, Greece, **8** Department of Medicine, Psychiatry and Dermatology, University of Zaragoza, Zaragoza, Spain

* bolivan@unizar.es

## Abstract

### Background

Chronic pain is a common complaint having distressing consequences for those that suffer from it. Pain and depression concur within the context of comorbidity, and both share underlying stress conditions. Sense of coherence (SOC) is a factor that determines how well an individual manages stress and stays healthy. Its relationship with depression has been frequently reported in the literature. Our objective was to assess the amount of evidence available regarding the association between SOC and depression in patients suffering from chronic pain.

### Methods

A systematic review and meta-analysis were performed. Searches were conducted between November 01 and December 31, 2020 in PubMed, Web of Science, Embase, PsycINFO, Psicodoc, ScienceDirect and Dialnet. There were no restrictions regarding the date of publication of the study. Evidence related to the relationship between SOC and depression in patients with chronic pain was summarized and compared.

### Results

A total of 163 articles were identified. We included 9 papers in the qualitative and quantitative synthesis. The pooled correlation coefficient was -0.55 (95%: -0.70; -0.41) and was not modified after removing any study. The heterogeneity across the studies was considerable ($I2 = 94.8\%$; $p < 0.001$). The random-effects meta-regression models for the association between SOC and depression showed that age ($p = 0.148$) and percentage of women ($p = 0.307$) were not related to heterogeneity across studies. No publication bias was detected ($p = 0.720$).

**Data Availability Statement:** All relevant data are within the paper and its Supporting Information files.

**Funding:** This work was supported by Carlos III Health Institute (ISCIII) grant number PI18/01336, the Aragonese Primary Care Research Group (GAIAP, B21_20R) that is part of the Department of Innovation, Research and University at the Government of Aragón (Spain) and the Institute for Health Research Aragón (IIS Aragón); the Research Network on Chronicity, Primary Care and Health Promotion (RICAPPS, RD21/0016/0005) that is part of the Results-Oriented Cooperative Research Networks in Health (RICORS) (Carlos III Health Institute); and Feder Funds "Another way to make Europe". The funders had no role in study design, data collection and analysis, decision to publish, or preparation of the manuscript.

**Competing interests:** The authors have declared that no competing interests exist.

## Conclusions

At first glance, the included studies give the impression that SOC is an important factor in depression levels of patients with chronic pain. Most of the included studies revealed a moderate association between SOC and depressive symptoms.

## Introduction

### Chronic pain

Chronic pain is a distressing complaint by its nature, which generates distressing consequences for those who suffer from it. It is defined as pain that persists or recurs for more than three months and can be experienced as a disease in itself (e.g., fibromyalgia or nonspecific low-back pain) or secondary to another disease (cancer, neuropathic pain, visceral pain, postsurgical pain, headache, orofacial and musculoskeletal pain) [1].

Prevalence estimates vary widely, even in studies of the same population. In a meta-analysis performed by Steingrímsdóttir et al. in 2017, prevalence estimates ranged from 8.7% to 64.4%, with a combined mean of 31% [2].

Psychosocial factors are an interactive complex of biopsychoecosocial processes that characterize chronic pain, but they are often granted a secondary status and are commonly considered to be reactions to pain. Observational research supports a strong bidirectional link between mood disorders (depression, anxiety, and distress) and persistent pain [3].

### Depression as comorbidity of patients with chronic pain

Depression is a common emotional disorder worldwide. The World Health Organization estimates that over 264 million people of all ages suffer from depression, with over 800,000 people committing suicide each year. And depression is a leading cause of disability worldwide [4]. No estimation of suicidal ideation, with no history of attempt, is possible. Depression requires the presence of symptoms such as depressed mood, sleep cycle disturbances, fatigue and poor concentration, which are different from normal reactive fluctuations in mood and emotional responses to challenges of daily life [5].

Pain and depression are comorbid conditions and their relationship has been studied extensively [6, 7]. Data reflect the importance of this comorbid relationship, since between 20 and 50% of all patients with chronic pain experience co-existent depression, and patients with severe pain are more likely to be depressed [8, 9]. Studies suggest that pain and depression may co-exacerbate physical and psychological symptoms and may prolong the duration of symptoms, leading to poor functional physical, mental and social outcomes compared to patients who have only depression or only pain [10]. The topic becomes more complex when additional variables, such as frailty are added to the 'equation'.

### Depression and sense of coherence (SOC)

Antonovsky (1996) developed the Salutogenesis theory, in which the sense of coherence (SOC) concept is the central hub. According to this theory, an individual's ability to modify his/her lifestyle is influenced by personal, interpersonal or contextual resources and SOC [11], which allows them to remain in reasonably good physical and emotional health despite stressors. SOC refers to an individual's ability to understand and effectively manage stressful stimuli derived from the environment, and is defined as: "a global orientation that expresses the extent to which one has a pervasive, enduring though dynamic feeling of confidence that (1) the stimuli deriving from one's internal and external environments in the course of living are

structured, predictable, and explicable (comprehensibility); (2) the resources are available to one to meet the demands posed by these stimuli (manageability); and (3) these demands are challenges, worthy of investment and engagement (meaningfulness)" [12].

SOC is a determinant factor in the development and maintenance of health since it influences how an individual manages stress and stays healthy [13]. Several studies have confirmed that SOC is inversely related to depression, presenting a protective capacity against depressive symptoms [14–17]. Higher SOC levels are associated with better mental health outcomes and may be a health-promoting resource [18, 19]. A significant inverse association has been found between the Beck Depression Inventory and SOC in a study from Crete (B-coef = -0.556, p < 0.001) [20]. The positive association between religious/spiritual beliefs and sense of coherence deserves further examination to promote a multidimensional approach in its study, as shown in another Cretan study [21]. This deserves attention as a potential explanatory mechanism regarding the association between depression and SOC [20], as it has been reported that religious and spiritual involvement may help patients to cope better with stressful situations by providing meaning and hope, and also may result in healthier lifestyle practices [22]. SOC can be measured with Antonovsky's 29-item Sense of Coherence Scale (SOC-29) [23]. It assesses an individual's view of life as being comprehensible, manageable and meaningful. The instrument consists of 29 items rated on a 7-point scale, scoring between 29 and 203 points. A shorter version of the scale exists, the SOC-13. It contains 13 items with Likert scales from 1 to 7, scoring between 13 and 91 points. The SOC-29 has consistency rates ranging from 0.82 to 0.95 and those of the SOC-13 range from .74 to .91. Higher scores indicate greater SOC [24].

## Depression and SOC in chronic pain patients

SOC is related to the patient´s ability to cope with stressors through an appropriate adoption of various coping strategies [23]. Individuals having a high SOC interpret stressors as being predictable, comprehensible, explicable and worth overcoming, and consider that they will overcome these stressors [12]. On the other hand, patients having a low SOC tend to display a poorer acceptance and adaptation to the illness and more depressive symptoms, poorer level of functioning and higher pain ratings [25–31].

Thus, SOC is found to relate to the choice of effective coping strategies that allow patients to adapt to chronic diseases, achieving an internal balance through a high SOC, while minimizing the consequences of these diseases, such as negative thoughts and depressive symptoms [14, 23, 29]. Individuals with stronger SOC take less medication, and assess pain as being less severe and having a lower effect on their mood. These individuals declared that they could control pain and have less often catastrophizing thoughts. Therefore, we can conclude that the role of SOC as a buffer in the functioning of sick individuals has been confirmed [32].

Some studies, however, have not revealed an association between SOC and depression, nonspecific physical symptoms or the overall health of patients with pain [33]. This suggests the need for prospective studies to clarify whether or not SOC is a predictor of psychosocial adaptation in acute and chronic disease [26].

Currently, no systematic reviews have been found that consider the association between SOC and depression in patients with chronic pain, nor have meta-analyses been created for literature quantifying this association. No other reviews focus on studies of cancer patients' caregivers, informal caregivers or mortality [18, 34, 35]. Therefore, in this review and meta-analysis, our objective is to expand the available evidence relating to SOC and depression in patients with chronic pain, to estimate the size of the association by reporting on the quality of the evidence. A secondary objective is to analyze this relationship in terms of age and gender, as a 'primordial' variable subset that is commonly examined within the context of biomedical

research. The hypothesis of this study is that SOC and depression are associated in patients with chronic pain, regardless of age and gender.

## Materials and methods

### Search strategy

A protocol using the Meta-analysis of Observational Studies in Epidemiology standards (MOOSE) [36] was completed before beginning the literature search (S1 File). In order to offer an overview of the existing literature on SOC and its relationship with depression, we searched for: depression and sense of coherence (and MeSH terms). Then, the articles were classified by theme, and those related to chronic pain morbidity were selected. The search strategy was performed in this manner, since searching with the term *chronic pain* resulted in too many articles with keywords that were nosologic disorders related to chronic pain and were left out. Searches were carried out on PubMed, Web of Science, Embase, PsycINFO, Psicodoc, ScienceDirect and Dialnet. Details are registered at PROSPERO- ID = CRD42020157512. Retrieved articles from all databases were referenced in Mendeley Reference Management Software (Version 1.19.4, Mendeley Ltd.). Searches were conducted from November 01 to December 31, 2020. All papers were screened by title and full text for possible inclusion, after discarding duplicates. Studies from the accurate search should fully meet the following inclusion criteria: Individuals suffering from depression, adults (>18 years old) and both sexes. There were no restrictions regarding the number of participants or the date of publication of the study. Studies with underage individuals, non-indexed articles, systematic reviews and meta-analyses, theoretical studies, conference articles, insufficient data, single case studies and articles not written in English or Spanish were excluded. Fig 1 offers an extensive and accurate search flow chart.

### Risk of bias

Considering the recommendations of Boyle (1998) [37] and Viswanathan et al. (2013) [38] and according to del-Pino-Casado et al. (2019) [39], we used the following criteria for assessing the methodological quality of the individual studies: (1) representative sampling (probabilistic sampling); (2) reliability and validity of measures: content validity and internal consistency in the target or similar population; (3) control of confounding factors; (4) for longitudinal studies: (4.1) follow-up of at least six months and (4.2) rate of follow-up of at least 80% of the original participating population. We considered confounders that were controlled for in the studies by taking into account those related to the design and/or analysis (i.e., matching, stratification, interaction terms, multivariate analysis, or other statistical adjustments such as instrumental variables) [40]. In cases of statistical adjustment, we considered that no confounding bias was present when the variation of the point estimate was less than 10% [41]. Three researchers (AAL, BOB, AAM) independently assessed the quality of the included studies. The studies' quality was assessed by pairs and any disagreements were resolved through discussion. Table 1 shows the quality methodology assessment for each selected study.

### Data analysis

Correlation coefficients were used as estimators of effect size and 95% confidence intervals (CIs) were established for each study. A pooled correlation coefficient was estimated using random-effects models based on the DerSimonian and Laird method (2007) [42]. In addition, we estimated the heterogeneity across studies using the $I^2$ statistic. Heterogeneity was considered as not important (0%–40%), moderate (30%–60%), substantial (50%–90%), or considerable (75%–100%) [43]. Moreover, the corresponding p-values were also taken into account.

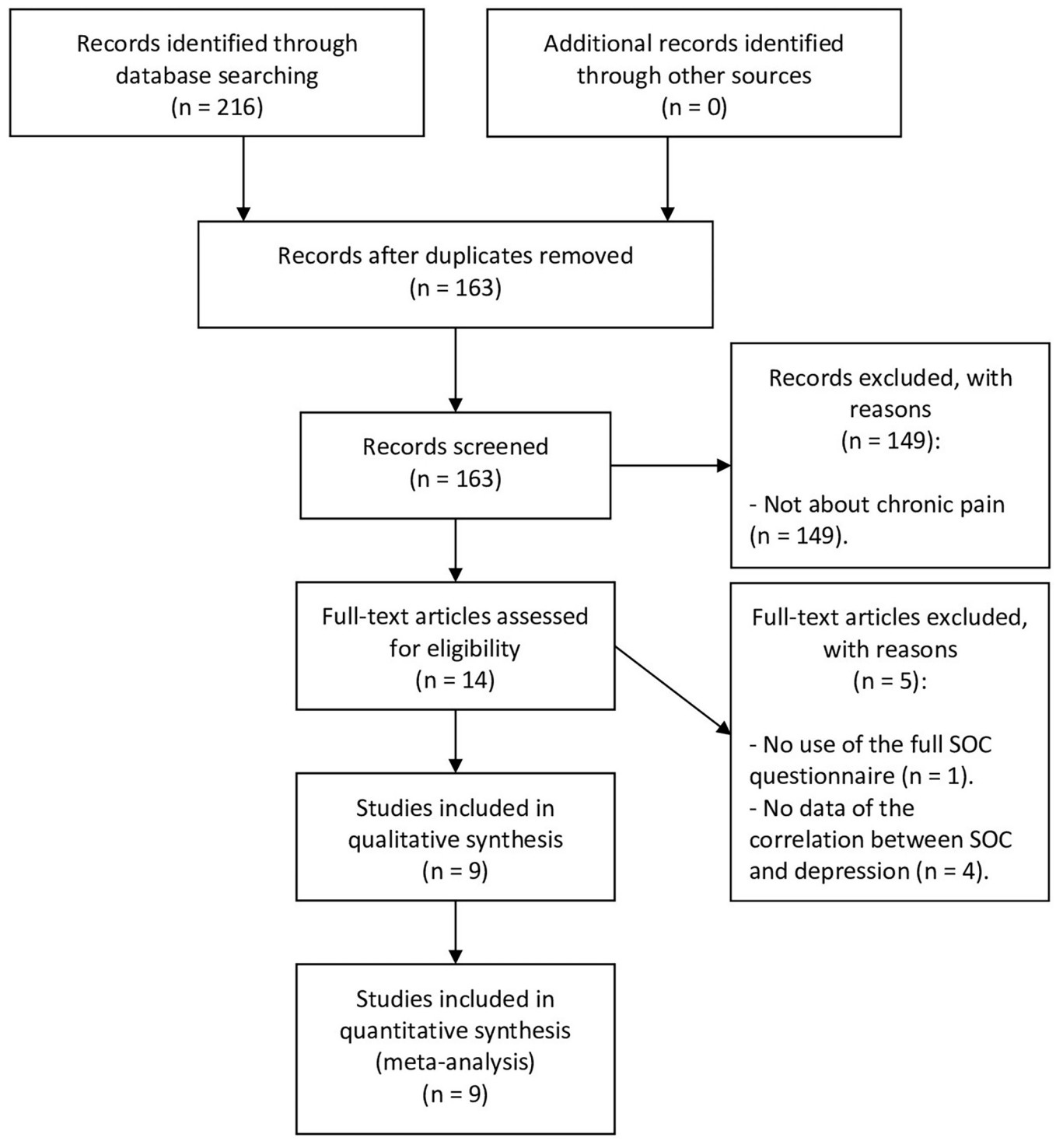

**Fig 1. Flow diagram for articles identified, screened eligible, and included in this paper.**

Sensitivity analysis was performed, individually excluding studies from the pooled estimate, to evaluate whether or not any study modified the original summary estimate. Random-effects meta-regressions were used to determine if the results were associated with participant's age

**Table 1. Methodological quality assessment of individual studies.**

| Study | C1 | C2 | C3 | C4.1 | C4.2 |
|---|---|---|---|---|---|
| **Bäck et al., 2019** | - | - | + | + | + |
| **Duvdevany et al., 2011** | - | + | + | NA | NA |
| **Hållstam et al., 2016** | - | - | + | + | + |
| **Hållstam et al., 2017** | - | - | + | + | + |
| **Nilsson et al., 2010** | + | + | + | NA | NA |
| **Pakarinen et al., 2017** | - | - | + | + | - |
| **Piekutin et al., 2018** | - | + | + | NA | NA |
| **Schnyder et al., 1999** | - | - | + | NA | NA |
| **Schrier et al., 2012** | - | - | + | NA | NA |

*Note.* C1: Representative sampling (probabilistic sampling); C2: reliability and validity of measures: content validity and internal consistency in the target or similar population; C3: control of confounding factors; C4: for longitudinal studies: C4.1: follow-up of at least six months and C4.2: rate of follow-up of at least 80% of the original participating population.

and percentage of women, since these variables may explain the observed heterogeneity. Finally, publication bias was estimated using Egger's test. STATA SE software (version 16, StataCorp LLC) [44], was used for the statistical analyses.

## Results

### Study flow and characteristics

A summary of the main results is presented in Table 2. The search strategy generated 216 papers of potential relevance to this study. After removing duplicates, 163 papers were screened. Then, only 14 papers remained, based on the eligibility criteria (individuals suffering from depression, adults (>18 years old), both sexes and having a chronic pain-related condition). Initially, 6 of them did not offer data on the correlation between SOC and depression and one of them did not use the full SOC questionnaire. Authors of articles that did not provide correlations were contacted by e-mail to request said data. Only one author (of two papers) responded to the request for data, therefore, the number of papers excluded due to missing data was reduced to 4. We included 9 papers in the qualitative and quantitative synthesis. One of these 9, included in the quantitative synthesis, has two different samples that were included separately in the analysis.

### Meta-analysis

The pooled ES values for the association between depression and SOC in patients suffering from chronic pain was -0.55 (95%: -0.70; -0.41). The heterogeneity between studies was considerable ($I^2$ = 94.8%; p < 0.001) (Fig 2).

In patients with temporomandibular disorders (TMD), having a lower Hospital Anxiety and Depression Scale subscale for depression (HADS-D) score and a higher SOC-13 score protected against the likelihood of exhibiting severe orofacial pain (r = -0.41) [25]. In patients with systemic lupus erythematosus (SLE), SOC was significantly associated with lower emotional distress (r = -0.67) [27]. In patients with non-malignant chronic pain, SOC was significantly associated with HADS-D (r = -0.61) [45] and (r = -0.67) [46]. In patients with temporomandibular disorder (TMD) pain, no statistically significant associations were found between mean SOC and levels of non-specific depression (r = -0.12) [33]. In patients with

**Table 2. Summary results from search strategies.**

| Study | Study design | Pathology | Sample (n/ women/ age) | Instruments | Correlation |
|---|---|---|---|---|---|
| **Bäck et al., 2019** | Prospective cohort study/Cross-sectional | Temporomandibular disorders (TMD) | n = 1059/ 100%/ middle-aged | SOC-13 HADS | r = -0.41 |
| **Duvdevany et al., 2011** | Cross-sectional | Systemic lupus erythematosus (SLE) | n = 100/ 88%/ 37 (11.8) | SOC-13 HADS | r = -0.57 |
| **Hållstam et al., 2016** | Prospective, observational study with a one-year follow-up | Non-malignant chronic pain | n = 35/ 90.5%/ 43.6 (15.7) | SOC-13 HADS | r = -0.61 |
| **Hållstam et al., 2017** | Longitudinal observational study | Different categories of chronic pain conditions | n = 235/ 64%/ 48 (37–62) | SOC-13 HADS | r = -0.67 |
| **Nilsson et al., 2010** | RCT | Temporomandibular disorder (TMD) pain. | Treatment group: n = 36/ 78%/ ns Control group: n = 37/ 86%/ ns | SOC-29 SCL-90-R | r = -0.12 |
| **Pakarinen et al., 2017** | Prospective clinical study | Lumbar spinal stenosis (LSS) | *Low SOC*: n = 32/ 65.6%/ 69.5 (12.8) *High SOC*: n = 42/ 64.3%/ 65.8 (9.6) | SOC-13 BDI | r = -0.37 |
| **Piekutin et al., 2018** | Cross-sectional | Ankylosing spondylitis (AS) | n = 82/ 11%/ 42 | SOC-29 BDI | r = -0.86 |
| **Schnyder et al., 1999 (Sample 1)** | Cross-sectional | Severe injuries following a life-threatening accident (SIAV) | Study 1: 89/ 73%/ 61.3 | SOC-29 HADS | r = -0.44 |
| **Schnyder et al., 1999 (Sample 2)** | Cross-sectional | Rheumatoid arthritis (RA) | Study 2: 112/ 25.9%/ 37.9 | SOC-29 HADS | r = -0.55 |
| **Schrier et al, 2012** | Cross-sectional | Nonmetastatic breast cancer | Study group: 40/ 100%/ 55.3 ± 7.3 Control group: 40/ 100%/ 53.5 ± 6.2 | SOC-13 HADS | r = -0.77 |

SD: standard deviation; HADS: Hospital Anxiety and Depression Scale; SOC: Sense of Coherence Scale; SCL-90-R: Symptom Checklist-90-Revised scale; BDI: Beck Depression Inventory Scale; SOC-L9: Leipzig short scale for recording the sense of coherence; BMQ: Berlin mood questionnaire; RCT: Randomized Control Trial.

lumbar spinal stenosis (LSS), a significant association was detected between a low SOC and a high Beck Depression Inventory (BDI) (r = -0.37) [28]. In patients with ankylosing spondylitis (AS), a very highly significant negative correlation was obtained between the overall SOC and the risk of depression (r = –0.857). Also, the lower the risk of depression, the higher the SOC [29]. In patients with severe injuries following a life-threatening accident (SIAV), significant negative correlations were found with the Derogatis Symptom Checklist, Revised (SCL-90-R) depression subscale score (r = -0.44) [26] and in patients with rheumatoid arthritis (RA), significant negative correlations were found between SOC and the HADS-D (r = -0.55) [26]. In patients with nonmetastatic breast cancer, a significant negative correlation was detected between the SOC scores and the parameters of affective condition (r = -0.77) [31].

## Sensitivity analysis

The sensitivity analysis revealed that the pooled ES estimation was not modified after removing any of the studies. It is shown in Table 3.

**Meta-regressions.** The random-effects meta-regression models for the association between SOC and depression showed that age (p = 0.148) and percentage of women (p = 0.307) were not related to heterogeneity across studies (Figs 3 and 4).

**Publication bias.** Publication bias was not detected for the estimation of the association between depression and SOC in patients with chronic pain (p = 0.720) (Fig 5).

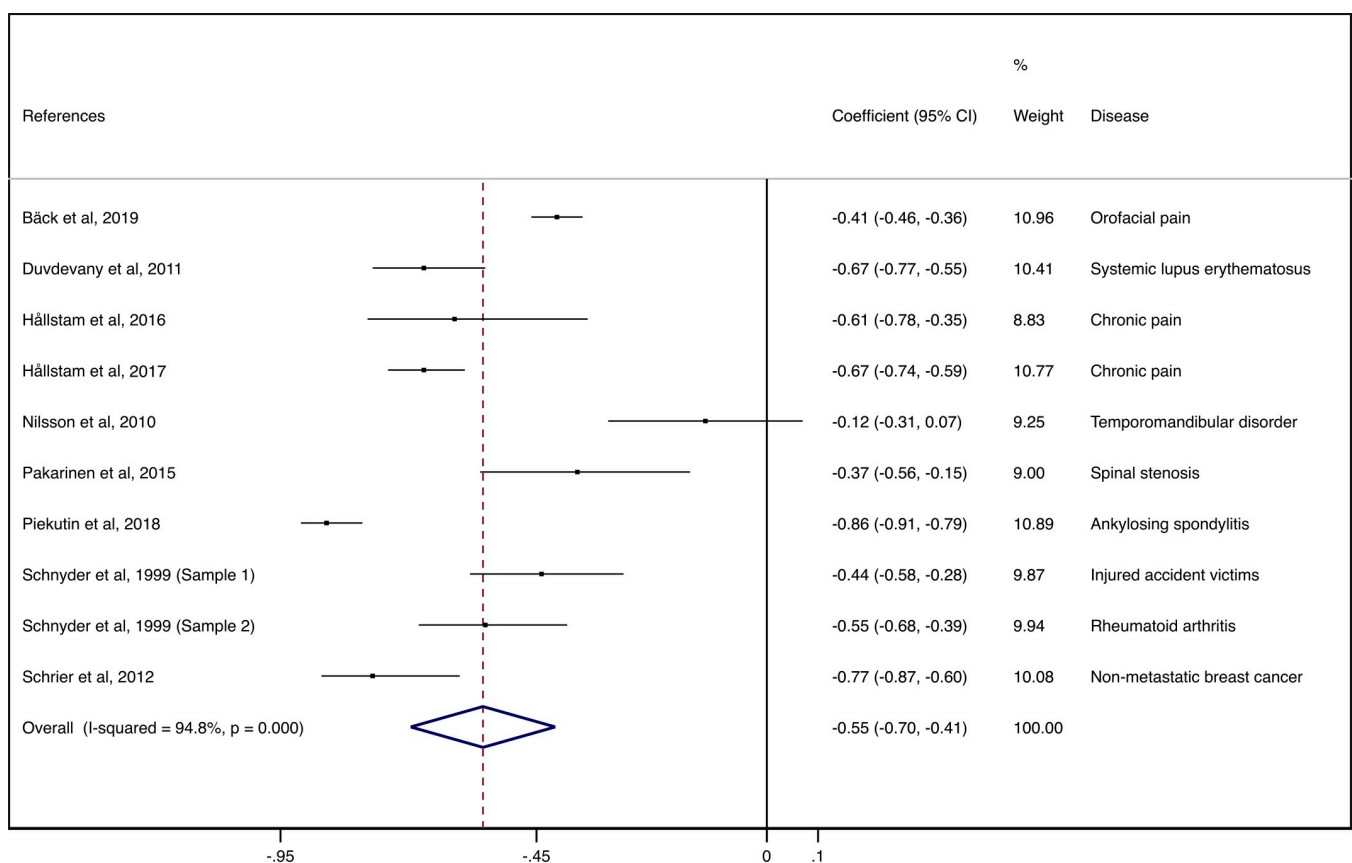

**Fig 2. Pooled ES values for the association between depression and sense of coherence in patients with chronic pain.**

## Discussion

Our systematic review and meta-analysis pooled the data from 9 research articles. To the best of our knowledge, this is the first meta-analysis to study the association between SOC and depression in patients suffering from chronic pain. As for the main objective of our review, findings revealed that this association is yet to be explored. In the selection process, 149 articles

**Table 3. Sensitivity analysis of the studies selected.**

| Study | ES | LL | UL |
|---|---|---|---|
| **Bäck et al, 2019** | -0.57 | -0.71 | -0.44 |
| **Duvdevany et al, 2011** | -0.54 | -0.70 | -0.38 |
| **Hållstam et al, 2016** | -0.55 | -0.70 | -0.40 |
| **Hållstam et al, 2017** | -0.54 | -0.70 | -0.37 |
| **Nilsson et al, 2010** | -0.60 | -0.74 | -0.46 |
| **Pakarinen et al, 2017** | -0.57 | -0.72 | -0.42 |
| **Piekutin et al, 2018** | -0.52 | -0.64 | -0.40 |
| **Schnyder et al, 1999 (Sample 1)** | -0.57 | -0.72 | -0.42 |
| **Schnyder et al, 1999 (Sample 2)** | -0.55 | -0.71 | -0.40 |
| **Schrier et al, 2012** | -0.53 | -0.68 | -0.38 |

*Note.* ES: Effect size; LL: Lower limit; UL: Upper limit.

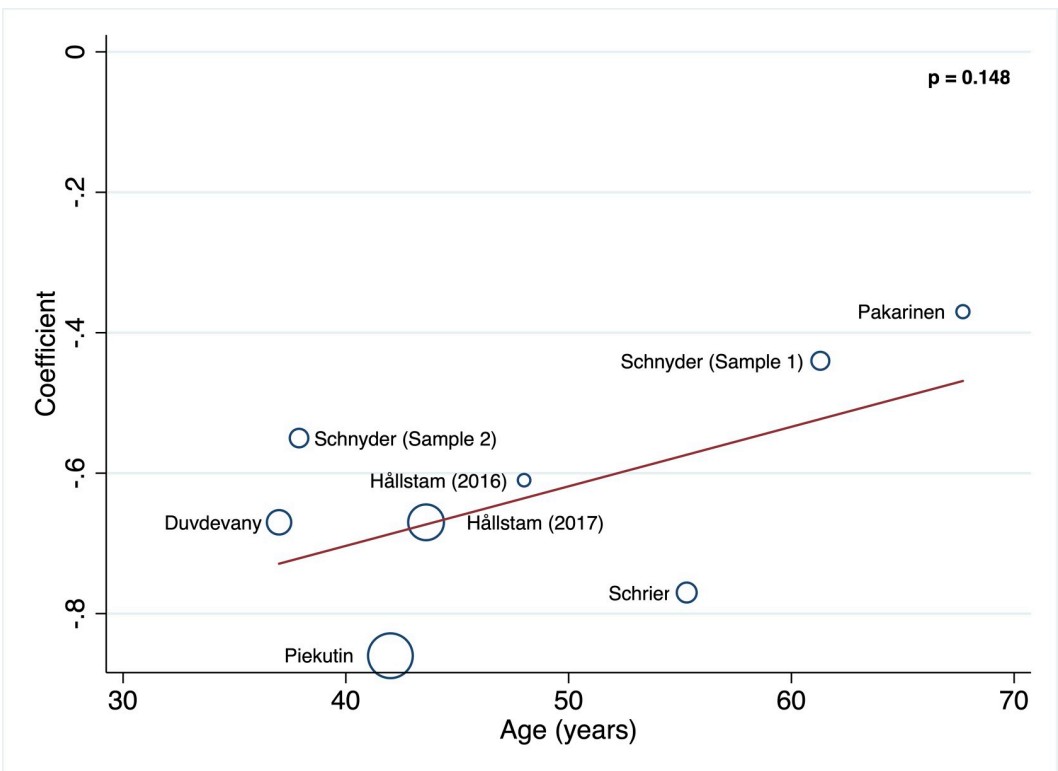

**Fig 3. Random-effects meta-regression models for the association between SOC and depression in age.**

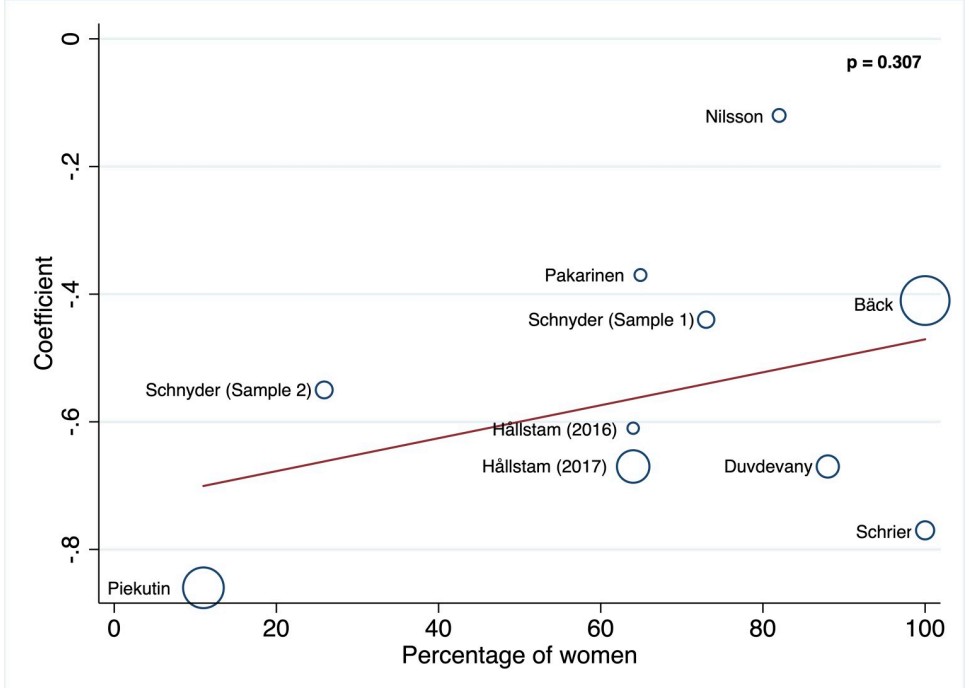

**Fig 4. Random-effects meta-regression models for the association between SOC and depression in gender.**

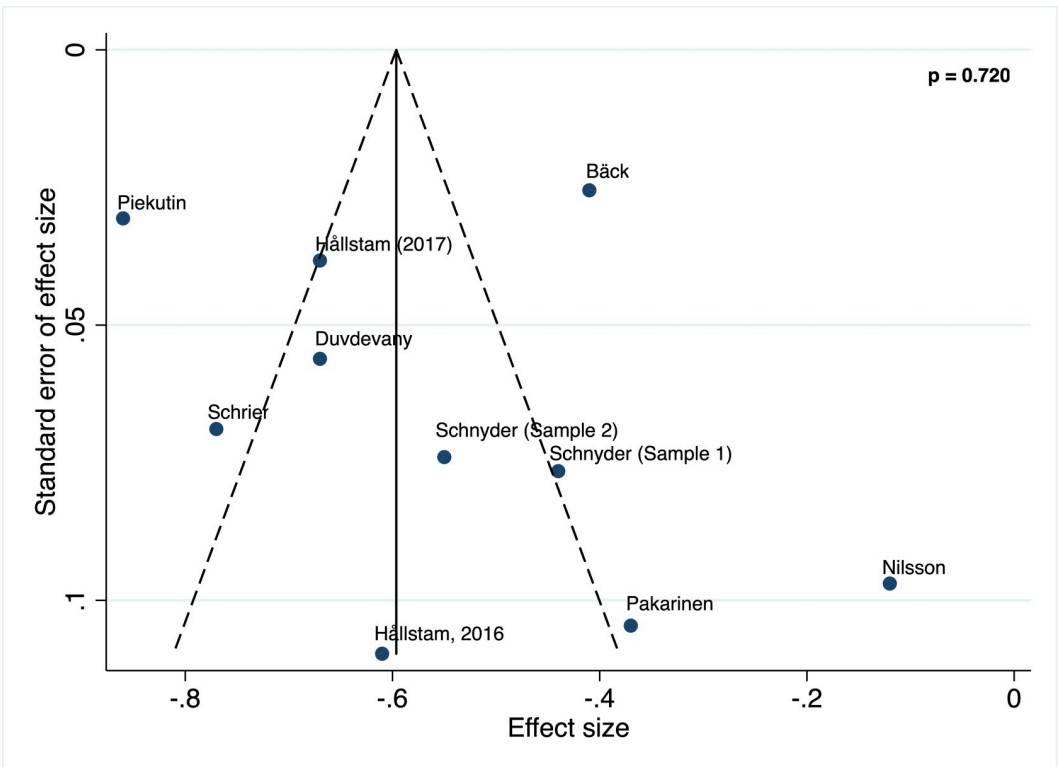

**Fig 5. Egger test for publication bias.**

were excluded, since they did not analyze the relationship between SOC and depression in chronic pain patients. Only 14 studies focused on chronic pain patients.

At first glance, the included studies indicate that SOC is an important factor in depression levels in chronic pain patients. The majority of the included studies reveal a negative association between SOC and depression symptoms (r = -0.55). Although the heterogeneity between studies was considerable, with the random-effects meta-regression models, we found that this heterogeneity was not due to differences in age and sex in the sample. Other studies have also found no differences regarding age, sex, ethnicity, nationality or study design [13]. Therefore, the heterogeneity reported in this metanalysis could be due to the variety of illnesses that cause chronic pain (temporomandibular disorders (TMD), systemic lupus erythematosus (SLE), non-malignant chronic pain, different categories of chronic pain conditions, lumbar spinal stenosis (LSS), ankylosing spondylitis (AS), severe injuries following a life-threatening accident (SIAV), rheumatoid arthritis (RA) and non-metastatic breast cancer).

These results are quite in line with previous research conducted in the general population, in individuals with chronic comorbidity; burnout and caregivers' burden; cancer; pregnancy and parenthood and neurodegenerative diseases. A quantitative study conducted in rural Crete (the SPILI III project; n = 195, mean age 67.2 ± 15.2) found that higher SOC scores are related to lower Beck Depression Inventory Scale (BDI) scores, indicating a lower frequency of depression in individuals with high SOC [20]. Similarly, the results for a Swedish cohort study of adolescent females with and without major depressive disorders (MDD) and anxiety disorders (n = 139, aged 14 to 18-year-old) led the authors to suggest that the SOC scale may be an inverse measure of depressive symptoms [47]. In a Finnish study (n = 4642, aged 25 to 74-year-old), a high inverse correlation was obtained between SOC and depression (r = -0.62).

Also, SOC was positively associated and depressive symptoms and anxiety were negatively associated with various health-related measures [48]. A study investigating SOC in community-dwelling older Spanish adults (n = 1106, aged 60 years and older) revealed that higher SOC was significantly associated with higher functional status and has a moderate correlation with depression (r = -0.47, p < 0.001) [49]. In a sample of 548 Japanese students (age range 18–37 years), SOC was negatively associated with mental health [50]. The results of a clinical trial with depressive patients suggest that both rehabilitation and conventional depression treatment in a first episode of depression may enhance the SOC and rehabilitation itself enhances SOC more effectively in those with less severe depression [51]. SOC has also been found to be a key psychological resource supporting the resilience of caregivers and their ability to cope with stressors. This in turn protects their psychological health. SOC remained relatively stable over a one-year period [52]. It also mediates factors related to postpartum depression (PPD), so that intervention for enhancing SOC is recommended for women at risk of PPD [53]. Definitely, the stronger the SOC, the fewer the symptoms of perceived depression [10].

Focusing on the research analyzed, it is found that chronic pain patients having a low level of SOC have more depressive symptoms, higher pain rates and a poorer level of functioning, less acceptance and less adaptation ability [25–31]. Only one study failed to uncover an association between SOC and depression [33]. These results are in line with the literature on SOC and chronic pain. High SOC is related to lower mean scores in pain intensity [54] and a weak SOC increases the likelihood of sustained pain and functional problems [55].

SOC is an important health-promoting resource that induces a positive perceived state of wellbeing [13]. Strong SOC also allows people to identify, use, and 'recycle' available resources and therefore, minimize feelings of tension and stress [56]. So, a strong SOC improves one's ability to deal with health problems [12, 57, 58]. Some findings provide initial evidence that SOC is a significant unique predictor of symptom change [59]. Individuals with a weak SOC appear to be less optimistic and display less learned resourcefulness, self-efficacy, hardiness, locus of control, mastery, self-esteem, acceptance of disability, and social skills [13]. SOC has a mediating effect on the influence of positive psychological constructs, life experiences, and social systems on our well-being [60].

Adopting a systematic salutogenic orientation to focus on available cognitive and emotional resources, strengthen them and create new ones could be very useful [12, 55]. The results encourage further clarification of the role and use of SOC in the rehabilitation context to minimize feelings of tension and improve function in everyday life [56]. Also, making individuals more resilient is of imperative need during the current pandemic situation and further research of the SOC issues in relation to depression seems to be a high priority.

## Limitations

A general limitation is the intrinsic flaws that meta-analytic studies contain [61]. We note that most of the studies identified in our review had cross-sectional or longitudinal designs [25–31, 45, 46]—excluding a Randomized Control Trial (RCT) [33]. Therefore, we had to work with correlation coefficients, and, as such, directional or causality relationships could not be determined. Additionally, the heterogeneous results imply that additional research is needed to see if other moderating variables, such as the sample's level of education [27, 55] and general health indicators of the sample [48], are important to the SOC-depression association. Besides this, SOC could be overlapped with other concepts with similar core dimensions, such as resilience, sense of coherence, hardiness, purpose in life, self-transcendence [62] or meaning in life [63]. Focusing only on one concept can result in missing other parallel concepts that have a certain significance and have an influence on therapeutic interventions.

Furthermore, a prior aim of this research was to examine the interaction between SOC and depression in the general population. This was not possible due to the variety of the samples and the lack of studies about the general population. As such, RCTs and longitudinal studies with general population samples that include the questionnaires in their measurements are needed. Furthermore, future research is needed to not only find out if there is a dominant influence of one variable on another but also clarify the relationship between SOC and chronic pain, and SOC and depression.

## Conclusions

Our findings provide evidence of the relationship between SOC and depression in individuals suffering from chronic pain and offer evidence of the importance of adequate levels of SOC. The perception of chronic pain, in terms of emotional triggering, caused from different disorders, deserves attention. However, the impact of direct SOC on depressive symptoms in chronic pain patients is unclear. Therefore, additional RCTs and longitudinal research studies are needed.

## Supporting information

**S1 File. MOOSE (Meta-analyses Of Observational Studies in Epidemiology) checklist.** (DOCX)

## Acknowledgments

We wish to thank the Aragonese Primary Care Research Group (GAIAP, B21_20R) that is part of the Department of Innovation, Research and University at the Government of Aragón (Spain) and the Institute for Health Research Aragón (IIS Aragón); the Research Network on Chronicity, Primary Care and Health Promotion (RICAPPS, RD21/0016/0005) that is part of the Results-Oriented Cooperative Research Networks in Health (RICORS) (Carlos III Health Institute).

## Author Contributions

**Conceptualization:** Alejandra Aguilar-Latorre, Bárbara Oliván-Blázquez, Christos Lionis, Emmanouil K. Symvoulakis, Rosa Magallón-Botaya.

**Data curation:** Alejandra Aguilar-Latorre, Ángela Asensio-Martínez, Bárbara Oliván-Blázquez.

**Formal analysis:** Alejandra Aguilar-Latorre, Ángela Asensio-Martínez, Bárbara Oliván-Blázquez, Iván Cavero-Redondo.

**Funding acquisition:** Bárbara Oliván-Blázquez.

**Investigation:** Alejandra Aguilar-Latorre, Ángela Asensio-Martínez, Bárbara Oliván-Blázquez, Celia Álvarez-Bueno.

**Methodology:** Alejandra Aguilar-Latorre, Ángela Asensio-Martínez, Bárbara Oliván-Blázquez, Celia Álvarez-Bueno, Iván Cavero-Redondo.

**Project administration:** Alejandra Aguilar-Latorre, Bárbara Oliván-Blázquez.

**Supervision:** Alejandra Aguilar-Latorre, Bárbara Oliván-Blázquez.

**Validation:** Bárbara Oliván-Blázquez.

**Visualization:** Alejandra Aguilar-Latorre, Bárbara Oliván-Blázquez.

**Writing – original draft:** Alejandra Aguilar-Latorre, Ángela Asensio-Martínez, Bárbara Oliván-Blázquez, Celia Álvarez-Bueno, Iván Cavero-Redondo.

**Writing – review & editing:** Christos Lionis, Emmanouil K. Symvoulakis, Rosa Magallón-Botaya.

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
