## [Decision Letter · Decision Letter 0]

19 Aug 2021

PONE-D-21-14267

Association between sense of coherence and depression in patients with chronic pain: a systematic review and meta-analysis.

PLOS ONE

Dear Dr. Oliván-Blázquez,

Thank you for submitting your manuscript to PLOS ONE. After careful consideration, we feel that it has merit but does not fully meet PLOS ONE’s publication criteria as it currently stands. Therefore, we invite you to submit a revised version of the manuscript that addresses the points raised during the review process.

Please respond to the reviewer comments regarding PROSPERO and clarify your hypotheses about the relationships between chronic pain, sense of coherence, and depression.

We look forward to receiving your revised manuscript.

Kind regards,

Lisa Susan Wieland

Academic Editor

PLOS ONE

"We wish to thank the Primary Care Prevention and Health Promotion Network (RedIAPP Health Institute Carlos III, Spain); Research Group B21_R17 of the Department of Science, University and the Knowledge Society of the Government of Aragon (Spain) and Feder Funds "Another way to make Europe".

"This work was supported by Carlos III Health Institute grant number PI18/01336, Feder Funds "Another way to make Europe". The funders had no role in study design, data collection and analysis, decision to publish, or preparation of the manuscript."

4. We note that this manuscript is a systematic review or meta-analysis; our author guidelines therefore require that you use PRISMA guidance to help improve reporting quality of this type of study. Please upload copies of the completed PRISMA checklist as Supporting Information with a file name “PRISMA checklist”

6. We noticed you have some minor occurrence of overlapping text with the following previous publication(s), which needs to be addressed:

- https://www.mdpi.com/2071-1050/11/4/1115/html

The text that needs to be addressed involves the Limitations section.

In your revision ensure you cite all your sources (including your own works), and quote or rephrase any duplicated text outside the methods section. Further consideration is dependent on these concerns being addressed.

Reviewers' comments:

Reviewer's Responses to Questions

**Comments to the Author**

1. Is the manuscript technically sound, and do the data support the conclusions?

Reviewer #1: Yes

Reviewer #2: No

2. Has the statistical analysis been performed appropriately and rigorously? 

Reviewer #1: Yes

Reviewer #2: Yes

3. Have the authors made all data underlying the findings in their manuscript fully available?

Reviewer #1: Yes

Reviewer #2: Yes

4. Is the manuscript presented in an intelligible fashion and written in standard English?

Reviewer #1: Yes

Reviewer #2: No

5. Review Comments to the Author

Reviewer #1: Thank you for the opportunity to review this work. It is interesting to see the potential relationship between sense of coherence and depression in those with chronic pain conditions. The concept of sense of coherence in relation to pain and depression is really interesting and significant to help refine interventions to assist these comorbid conditions of chronic pain and depression. The authors described the process and significance of this work very clearly and thoroughly.

One curiosity I had as I read this work was the overlap between the concepts of sense of coherence, resilience, and spiritual wellbeing research—particularly the body of research on meaning and purpose. I appreciated the authors inclusion of spirituality as a mechanism and contributor to sense of coherence, as well as resilience. However, I do wonder if a potential limitation in this review and meta-analysis has to do with these overlapping concepts of sense of coherence, resilience, and meaning and purpose. Due to their overlapping nature, the research for each concept seems to be siloed so that there might be the possibility of researchers/clinicians speaking about parallel concepts and not understanding the relationships and significance to each other and to therapeutic interventions.

Reviewer #2: The authors referenced a PROSPERO ID that described a systematic review of sense of "coherence and depression" from 2010 to 2020. This does not agree with the described protocol described in the manuscript that examines SOC and depression AND chronic pain. The authors should provide explicit descriptions of their inclusion/exclusion criteria for their review so that their findings can be replicated and confirmed.

Is SOC really an important factor in depression levels in people with chronic pain? The authors described/assessed the relationship between SOC and depression within the context of chronic pain. However, they provide no definitive data on this relationship. The authors describe the bidirectional nature of depression and chronic pain but make no mention of which variable has a dominant influence (if it exists) over the other. Perhaps chronic pain leads to lower sense of coherence and the relationship between depression and SOC is a byproduct of this. Perhaps then the authors can clarify further their interest/focus on the relationship between SOC and depression and not SOC and chronic pain with depression as a consequence of chronic pain or a consequence of lower sense of coherence.

6. PLOS authors have the option to publish the peer review history of their article (what does this mean?). If published, this will include your full peer review and any attached files.

Reviewer #1: **Yes: **Marlysa Sullivan DPT CIAYT

Reviewer #2: No

---

## [Author Response · Author response to Decision Letter 0]

31 Aug 2021

PONE-D-21-14267

Association between sense of coherence and depression in patients with

chronic pain: a systematic review and meta-analysis.

PLOS ONE

1. Please ensure that your manuscript meets PLOS ONE's style

requirements, including those for file naming. The PLOS ONE style

templates can be found at

and

Thank you for the information. It’s checked and done.

2. Thank you for stating the following in the Acknowledgments Section of

your manuscript:

"We wish to thank the Primary Care Prevention and Health Promotion

Network (RedIAPP Health Institute Carlos III, Spain); Research Group

B21_R17 of the Department of Science, University and the Knowledge

Society of the Government of Aragon (Spain) and Feder Funds "Another way

to make Europe".

We note that you have provided funding information that is not currently

declared in your Funding Statement. However, funding information should

not appear in the Acknowledgments section or other areas of your

manuscript. We will only publish funding information present in the

Funding Statement section of the online submission form.

Please remove any funding-related text from the manuscript and let us

know how you would like to update your Funding Statement. Currently,

your Funding Statement reads as follows:

"This work was supported by Carlos III Health Institute grant number

PI18/01336, Feder Funds "Another way to make Europe". The funders had no

role in study design, data collection and analysis, decision to publish,

or preparation of the manuscript."

Please include your amended statements within your cover letter; we will

change the online submission form on your behalf

Thank you. We have eliminated the funding information of the “Acknowledgments” section. Now it is stated as follows:

We wish to thank the Primary Care Prevention and Health Promotion Network (RedIAPP Health Institute Carlos III, Spain) and Research Group B21_R17 of the Department of Science, University and the Knowledge Society of the Government of Aragon (Spain).

3. PLOS requires an ORCID iD for the corresponding author in Editorial

Manager on papers submitted after December 6th, 2016. Please ensure that

you have an ORCID iD and that it is validated in Editorial Manager. To

do this, go to 'Update my Information' (in the upper left-hand corner of

the main menu), and click on the Fetch/Validate link next to the ORCID

field. This will take you to the ORCID site and allow you to create a

new iD or authenticate a pre-existing iD in Editorial Manager. Please

see the following video for instructions on linking an ORCID iD to your

Editorial Manager account: https://www.youtube.com/watch?v=_xcclfuvtxQ

Thank you. We cannot do it. The warning message stated: “there is already an identical orcid registered”. It is because she has another account with another email. Her ORCID is 0000-0001-6565-9699. We wrote you asking if it is possible to unify the two accounts. We are still waiting for the response. Thank you in advance.

4. We note that this manuscript is a systematic review or meta-analysis;

our author guidelines therefore require that you use PRISMA guidance to

help improve reporting quality of this type of study. Please upload

copies of the completed PRISMA checklist as Supporting Information with

a file name "PRISMA checklist"

Our research is a Meta analysis of observational studies, so we have completed the MOOSE (Meta-analyses Of Observational Studies in Epidemiology) Checklist, as it is stablished and cited in the paper (Supporting information - S1 File. MOOSE (Meta-analyses Of Observational Studies in Epidemiology) Checklist.).

5. Please include captions for your Supporting Information files at the

end of your manuscript, and update any in-text citations to match

accordingly. Please see our Supporting Information guidelines for more

information: http://journals.plos.org/plosone/s/supporting-information.

Thank you. Now it is done.

6. We noticed you have some minor occurrence of overlapping text with

the following previous publication(s), which needs to be addressed:

- https://www.mdpi.com/2071-1050/11/4/1115/html

The text that needs to be addressed involves the Limitations section.

In your revision ensure you cite all your sources (including your own

works), and quote or rephrase any duplicated text outside the methods

section. Further consideration is dependent on these concerns being

addressed.

Thank you. We have rewritten the entire paragraph.

Response to Reviewers

Reviewer #1: Thank you for the opportunity to review this work. It is interesting to see the potential relationship between sense of coherence and depression in those with chronic pain conditions. The concept of sense of coherence in relation to pain and depression is really interesting and significant to help refine interventions to assist these comorbid conditions of chronic pain and depression. The authors described the process and significance of this work very clearly and thoroughly. One curiosity I had as I read this work was the overlap between the concepts of sense of coherence, resilience, and spiritual wellbeing research--particularly the body of research on meaning and purpose. I appreciated the authors inclusion of spirituality as a mechanism and contributor to sense of coherence, as well as resilience. However, I do wonder if a potential limitation in this review and meta-analysis has to do with these overlapping concepts of sense of coherence, resilience, and meaning and purpose. Due to their overlapping nature, the research for each concept seems to be siloed so that there might be the possibility of researchers/clinicians speaking about parallel concepts and not understanding the relationships and significance to each other and to therapeutic interventions.

Answer reviewer 1: Thank you very much for your thorough review. We are very grateful for your comments and feedback. We would like to note that with the study being a systematic review and meta-analysis about two very specific concepts and questionnaires (Sense of Coherence, SOC – measured specifically by SOC-13 and SOC-29 – and depression), we needed to put aside the related concepts that you mentioned and narrow the search. Although we looked into it, finally, we chose to work with SOC as it is the core concept in the salutogenesis theory developed by Antonovsky (1996). This was due to its relevance, as well as the fact that the SOC questionnaire has been widely used and translated into at least 33 languages, and also seems to be a reliable, valid, and cross-culturally applicable instrument that measures how people manage stressful situations and stay well (with these three domains: comprehensibility, manageability, and meaningfulness). We also wrote about possible therapies that increase SOC, however, of course, these methods can also serve to increase resilience or other concepts with similar functions. We agree with you and recognize that in practice these concepts overlap. As you have said, relying only on one concept at a time, separately, can be a limitation.

As such, we have found more literature in this regard and added your compelling suggestion in the “Introduction” and “Limitations” sections:

INTRODUCTION

“The positive association between religious/spiritual beliefs and sense of coherence deserves further examination to promote a multidimensional approach in its study, as shown in another Cretan study (Stefanaki et al., 2014). This deserves attention as a potential explanatory mechanism regarding the association between depression and SOC (Anyfantakis et al., 2015), as it has been reported that religious and spiritual involvement may help patients to cope better with stressful situations by providing meaning and hope, and also may result in healthier lifestyle practices (Bonelli et al., 2012).”

LIMITATIONS

“A general limitation is the intrinsic flaws that meta-analytic studies contain (Rosenthal & DiMatteo, 2001). We note that most of the studies identified in our review had cross-sectional or longitudinal designs (Bäck et al., 2019; Duvdevany et al., 2011; Hållstam et al., 2016, 2017; Pakarinen et al., 2015; Piekutin et al., 2018; Schnyder et al., 1999, 2000; Schrier et al., 2012) — excluding a Randomized Control Trial (RCT) (Nilsson & Ekberg, 2010). Therefore, we had to work with correlation coefficients, and, as such, directional or causality relationships could not be determined. Additionally, the heterogeneous results imply that additional research is needed to see if other moderating variables, such as the sample's level of education (Duvdevany et al., 2011; Lillefjell et al., 2015) and general health indicators of the sample (Konttinen et al., 2008), are important to the SOC-depression association. Besides this, SOC could be overlapped with other concepts with similar core dimensions, such as resilience, sense of coherence, hardiness, purpose in life, self-transcendence (Lundman et al., 2010), or meaning in life (Bartrés-Faz et al., 2018). Focusing only on one concept can result in missing other parallel concepts that have a certain significance and have an influence on therapeutic interventions.

Furthermore, a prior aim of this research was to examine the interaction between SOC and depression in the general population. This was not possible due to the variety of the samples and the lack of studies about the general population. As such, RCTs and longitudinal studies with general population samples that include the questionnaires in their measurements are needed. Furthermore, future research is needed to not only find out if there is a dominant influence of one variable on another but also clarify the relationship between SOC and chronic pain, and SOC and depression.”

Reviewer #2: The authors referenced a PROSPERO ID that described a systematic review of sense of "coherence and depression" from 2010 to 2020. This does not agree with the described protocol described in the manuscript that examines SOC and depression AND chronic pain. The authors should provide explicit descriptions of their inclusion/exclusion criteria for their review so that their findings can be replicated and confirmed.

Is SOC really an important factor in depression levels in people with chronic pain? The authors described/assessed the relationship between SOC and depression within the context of chronic pain. However, they provide no definitive data on this relationship. The authors describe the bidirectional nature of depression and chronic pain but make no mention of which variable has a dominant influence (if it exists) over the other. Perhaps chronic pain leads to lower sense of coherence and the relationship between depression and SOC is a by product of this. Perhaps then the authors can clarify further their interest/focus on the relationship between SOC and depression and not SOC and chronic pain with depression as a consequence of chronic pain or a consequence of lower sense of coherence.

Answer reviewer 2: Thank you very much for your thorough review. We are very grateful for your comments and feedback. First of all, we agree with you and have consequently started an update in PROSPERO, as we did not update the information there due to the fact that we were extremely focused on the paper. When the PROSPERO managers approve the new version, it will be exactly the same as in the paper. Thank you for advising us.

Regarding your question, the analysis of SOC and depression in patients with chronic pain is due to our interest in analyzing the relationship between SOC and depression, as this paper is part of a bigger RCT research project in which we assess both variables. However, given the heterogeneity of the articles found, and the variability in the etiology of depression, it was decided to analyze this relationship in chronic pain patients, as its study is more frequent. Moreover, the relationship between SOC and depression has been highly reported comparing the relationship of SOC and chronic pain (for example, several studies have confirmed that SOC is inversely related to depression, presenting a protective capacity against depressive symptoms (López-Martínez et al., 2019; Plata-Muñoz et al., 2004; Skärsäter et al., 2009). 

As such, we have found more literature in this regard and added more information in the “Introduction” section, which we have taken from this paragraph:

“Individuals with stronger SOC take less medication, and assess pain as being less severe and having a lower effect on their mood. These individuals declared that they could control pain and have less often catastrophizing thoughts. Therefore, we can conclude that the role of SOC as a buffer in the functioning of sick individuals has been confirmed (Andruszkiewicz et al., 2017).”

However, as you have said, relying only on one specific relationship and in a definite direction can be a limitation. As such, we have rewritten the “Limitations” section:

“A general limitation is the intrinsic flaws that meta-analytic studies contain (Rosenthal & DiMatteo, 2001). We note that most of the studies identified in our review had cross-sectional or longitudinal designs (Bäck et al., 2019; Duvdevany et al., 2011; Hållstam et al., 2016, 2017; Pakarinen et al., 2015; Piekutin et al., 2018; Schnyder et al., 1999, 2000; Schrier et al., 2012) — excluding a Randomized Control Trial (RCT) (Nilsson & Ekberg, 2010). Therefore, we had to work with correlation coefficients, and, as such, directional or causality relationships could not be determined. Additionally, the heterogeneous results imply that additional research is needed to see if other moderating variables, such as the sample's level of education (Duvdevany et al., 2011; Lillefjell et al., 2015) and general health indicators of the sample (Konttinen et al., 2008), are important to the SOC-depression association. Besides this, SOC could be overlapped with other concepts with similar core dimensions, such as resilience, sense of coherence, hardiness, purpose in life, self-transcendence (Lundman et al., 2010), or meaning in life (Bartrés-Faz et al., 2018). Focusing only on one concept can result in missing other parallel concepts that have a certain significance and have an influence on therapeutic interventions.

Furthermore, a prior aim of this research was to examine the interaction between SOC and depression in the general population. This was not possible due to the variety of the samples and the lack of studies about the general population. As such, RCTs and longitudinal studies with general population samples that include the questionnaires in their measurements are needed. Furthermore, future research is needed to not only find out if there is a dominant influence of one variable on another but also clarify the relationship between SOC and chronic pain, and SOC and depression.”

While revising your submission, please upload your figure files to the

Preflight Analysis and Conversion Engine (PACE) digital diagnostic tool,

https://pacev2.apexcovantage.com/. PACE helps ensure that figures meet

PLOS requirements. To use PACE, you must first register as a user.

Registration is free. Then, login and navigate to the UPLOAD tab, where

you will find detailed instructions on how to use the tool. If you

encounter any issues or have any questions when using PACE, please email

PLOS at figures@plos.org. Please note that Supporting Information files

do not need this step.

---

## [Decision Letter · Decision Letter 1]

18 Nov 2022

PONE-D-21-14267R1Association between sense of coherence and depression in patients with chronic pain: a systematic review and meta-analysis.PLOS ONE

Dear Dr. Oliván-Blázquez,

Thank you for submitting your manuscript to PLOS ONE. After careful consideration, we feel that it has merit but does not fully meet PLOS ONE’s publication criteria as it currently stands. Therefore, we invite you to submit a revised version of the manuscript that addresses the points raised during the review process.

The manuscript has been evaluated by three reviewers, and their comments are available below.Although two of the reviewers are happy with the revised manuscript, the third reviewer has raised a number of concerns that need attention, including requests for additional information on methodological aspects of the study. Could you please revise the manuscript to carefully address the concerns raised?

We look forward to receiving your revised manuscript.

Kind regards,

Steve Zimmerman, PhD

Associate Editor, PLOS ONE

Reviewers' comments:

Reviewer's Responses to Questions

**Comments to the Author**

1. If the authors have adequately addressed your comments raised in a previous round of review and you feel that this manuscript is now acceptable for publication, you may indicate that here to bypass the “Comments to the Author” section, enter your conflict of interest statement in the “Confidential to Editor” section, and submit your "Accept" recommendation.

Reviewer #1: All comments have been addressed

Reviewer #2: All comments have been addressed

Reviewer #3: (No Response)

2. Is the manuscript technically sound, and do the data support the conclusions?

Reviewer #1: Yes

Reviewer #2: Yes

Reviewer #3: No

3. Has the statistical analysis been performed appropriately and rigorously? 

Reviewer #1: Yes

Reviewer #2: Yes

Reviewer #3: Yes

4. Have the authors made all data underlying the findings in their manuscript fully available?

Reviewer #1: Yes

Reviewer #2: Yes

Reviewer #3: No

5. Is the manuscript presented in an intelligible fashion and written in standard English?

Reviewer #1: Yes

Reviewer #2: Yes

Reviewer #3: Yes

6. Review Comments to the Author

Reviewer #1: Thank you for your response to my questions and the additions to the manuscript. I think this helps provide greater context and significance to this work

Reviewer #2: The authors are to be commended on this important undertaking. The PROSPERO information needs to be updated prior to publication.

Reviewer #3: Thank you for the opportunity to review this work. The work focused on the association between sense of coherence and depression in patients with chronic pain. It is interesting and valuable. However, the manuscript still has the following problems worthy of attention, through the improvement of these problems can better improve the quality of the manuscript.

Abstract:

1The author should describe the publication time of the included literature and the search time of this study respectively. The current statement is easy to cause ambiguity.

2The authors do not search the Cochrane Library (CENTRAL) database, but this database is also important for meta-analysis.

Introduction

1It is necessary to make appropriate research hypothesis according to the current research situation.

Methods

1The authors lack the necessary description of the method content. Although they have provided the registration number (PROSPERO- ID = CRD42020157512), it seems that the current system state does not provide key information about the method. Even if the authors use the PROSPERO system in their research, the manuscript should also provide the readers with necessary information for reference.

2Similarly, the manuscript lacks the description of the publication time of the included literature, the criteria for literature screening and other necessary information.

3The format of Table 1 needs to be modified. The three--line table should be better.

4The literature management software and data analysis software mentioned in the manuscript lack the necessary manufacturer information.

Results

The format of Table 3 needs to be modified.

Discussions

The results in the manuscript are limited, but the authors seem to have discussed too much extraneous content. Some of the discussions lack the support of the results, more like some inferences. It is suggested to narrow the discussion and focus on the key factor of SOC.

7. PLOS authors have the option to publish the peer review history of their article (what does this mean?). If published, this will include your full peer review and any attached files.

Reviewer #1: **Yes: **Marlysa Sullivan

Reviewer #2: **Yes: **Joel Alcantara, DC, PhD (candidate)

Reviewer #3: No

---

## [Author Response · Author response to Decision Letter 1]

1 Dec 2022

Comments to the Author

1. If the authors have adequately addressed your comments raised in a previous round of review and you feel that this manuscript is now acceptable for publication, you may indicate that here to bypass the “Comments to the Author” section, enter your conflict of interest statement in the “Confidential to Editor” section, and submit your "Accept" recommendation.

Reviewer #1: All comments have been addressed

Reviewer #2: All comments have been addressed

Reviewer #3: (No Response)

2. Is the manuscript technically sound, and do the data support the conclusions?

Reviewer #1: Yes

Reviewer #2: Yes

Reviewer #3: No

3. Has the statistical analysis been performed appropriately and rigorously?

Reviewer #1: Yes

Reviewer #2: Yes

Reviewer #3: Yes

4. Have the authors made all data underlying the findings in their manuscript fully available?

Reviewer #1: Yes

Reviewer #2: Yes

Reviewer #3: No

5. Is the manuscript presented in an intelligible fashion and written in standard English?

Reviewer #1: Yes

Reviewer #2: Yes

Reviewer #3: Yes

6. Review Comments to the Author

Reviewer #1: Thank you for your response to my questions and the additions to the manuscript. I think this helps provide greater context and significance to this work.

Answer reviewer 1: Thank you very much for your thorough review. We are very grateful for your comments and feedback.

Reviewer #2: The authors are to be commended on this important undertaking. The PROSPERO information needs to be updated prior to publication.

Answer reviewer 2: Thank you very much for your thorough review. We are very grateful for your comments and feedback. PROSPERO information has been updated.

Reviewer #3: Thank you for the opportunity to review this work. The work focused on the association between sense of coherence and depression in patients with chronic pain. It is interesting and valuable. However, the manuscript still has the following problems worthy of attention, through the improvement of these problems can better improve the quality of the manuscript.

Answer reviewer 3: Thank you very much for your thorough review. We are very grateful for your comments and feedback. Following your comments and suggestions, we have made changes to the manuscript, which we detail below, point by point.

Abstract:

1) The author should describe the publication time of the included literature and the search time of this study respectively. The current statement is easy to cause ambiguity.

Answer reviewer 3: Thank you. There were no restrictions regarding the publication date of the study since we knew that we would find few articles in general, since the topic was new. So, we have put this data in the abstract and in methodology since, as you say, it was not written. We have added the following sentence:

“There were no restrictions regarding the date of publication of the study.”

2) The authors do not search the Cochrane Library (CENTRAL) database, but this database is also important for meta-analysis.

Answer reviewer 3: Thank you. Our research is a Meta analysis of observational studies. And, as we put in the protocol: The searches were carried out in PubMed, Web of Science, Embase, PsycINFO, Psicodoc, ScienceDirect and Dialnet. We did not use the Cochrane Library (CENTRAL) database because we did not want to include other meta-analyses in our search; since, as we also put in the protocol: Studies with underage individuals, non-indexed articles, systematic reviews and meta-analyses, theoretical studies, conference articles, insufficient data, single case studies and articles not written in English or Spanish were excluded. In any case, following your suggestion, we have done a quick search to see what Cochrane results came out and we have not found anything relevant that we had not already taken into account. If you see it appropriate, we can add this search in methods. We are open to suggestions regarding this topic.

Introduction

1) It is necessary to make appropriate research hypothesis according to the current research situation.

Answer reviewer 3: Thank you. It is true that we have forgotten to add the hypothesis. We have already added it like this:

“The hypothesis of this study is that SOC and depression are associated in patients with chronic pain, regardless of their age and gender.”

Methods

1) The authors lack the necessary description of the method content. Although they have provided the registration number (PROSPERO- ID = CRD42020157512), it seems that the current system state does not provide key information about the method. Even if the authors use the PROSPERO system in their research, the manuscript should also provide the readers with necessary information for reference.

Answer reviewer 3: Thank you. We have reviewed what was written in PROSPERO and we have made sure that all the information written in PROSPERO is reflected in the manuscript. In addition, we have added the missing information as we have pointed out in the previous comments. If you think there is any more information missing about the methods, you can point it out, we are open to new suggestions.

2) Similarly, the manuscript lacks the description of the publication time of the included literature, the criteria for literature screening and other necessary information.

Answer reviewer 3: Thank you. As we have said in the previous comment, we have reviewed what was written in PROSPERO and we have made sure that all the information written in PROSPERO is reflected in the manuscript. In addition, we have added the missing information as we have pointed out in the previous comments. For example, we have added the following sentence:

“There were no restrictions regarding the date of publication of the study.”

Now, the paragraph about the "Search strategy" has this information:

“A protocol using the Meta-analysis of Observational Studies in Epidemiology standards (MOOSE) (34) was completed before beginning the literature search (S1). In order to offer an overview of the existing literature on SOC and its relationship with depression, we searched for: depression and sense of coherence (and MeSH terms). Then, the articles were classified by theme, and those related to chronic pain morbidity were selected. The search strategy was performed in this manner, since searching with the term chronic pain resulted in too many articles with keywords that were nosologic disorders related to chronic pain and were left out. Searches were carried out on PubMed, Web of Science, Embase, PsycINFO, Psicodoc, ScienceDirect and Dialnet. Details are registered at PROSPERO- ID = CRD42020157512. Retrieved articles from all databases were referenced in Mendeley Reference Management Software (Version 1.19.4, Mendeley Ltd.). Searches were conducted from November 01 to December 31, 2020. All papers were screened by title and full text for possible inclusion, after discarding duplicates. Studies from the accurate search should fully meet the following inclusion criteria: Individuals suffering from depression, adults (>18 years old) and both sexes. There were no restrictions regarding the number of participants or the date of publication of the study. Studies with underage individuals, non-indexed articles, systematic reviews and meta-analyses, theoretical studies, conference articles, insufficient data, single case studies and articles not written in English or Spanish were excluded. Fig 1 offers an extensive and accurate search flow chart.”

3) The format of Table 1 needs to be modified. The three--line table should be better.

Answer reviewer 3: Thank you. We have modified the format of the tables. We are open to any other modification that you consider appropriate.

4) The literature management software and data analysis software mentioned in the manuscript lack the necessary manufacturer information.

Answer reviewer 3: Thank you. We have added the following information:

“Mendeley Reference Management Software (Version 1.19.4, Mendeley Ltd.).” and “STATA SE software (version 16, StataCorp LLC)”.

Results

The format of Table 3 needs to be modified.

Answer reviewer 3: Thank you. We have modified the format of the tables. We are open to any other modification that you consider appropriate.

Discussions

The results in the manuscript are limited, but the authors seem to have discussed too much extraneous content. Some of the discussions lack the support of the results, more like some inferences. It is suggested to narrow the discussion and focus on the key factor of SOC.

Answer reviewer 3: Thank you. We have narrowed the discussion and we have eliminated much extraneous content, as suggested. Now the discussion is as follows: 

"Our systematic review and meta-analysis pooled the data from 9 research articles. To the best of our knowledge, this is the first meta-analysis to study the association between SOC and depression in patients suffering from chronic pain. As for the main objective of our review, findings revealed that this association is yet to be explored. In the selection process, 149 articles were excluded, since they did not analyze the relationship between SOC and depression in chronic pain patients. Only 14 studies focused on chronic pain patients.

At first glance, the included studies indicate that SOC is an important factor in depression levels in chronic pain patients. The majority of the included studies reveal a negative association between SOC and depression symptoms (r = -0.55). Although the heterogeneity between studies was considerable, with the random‐effects meta‐regression models, we found that this heterogeneity was not due to differences in age and sex in the sample. Other studies have also found no differences regarding age, sex, ethnicity, nationality or study design (13). Therefore, the heterogeneity reported in this metanalysis could be due to the variety of illnesses that cause chronic pain (temporomandibular disorders (TMD), systemic lupus erythematosus (SLE), non-malignant chronic pain, different categories of chronic pain conditions, lumbar spinal stenosis (LSS), ankylosing spondylitis (AS), severe injuries following a life-threatening accident (SIAV), rheumatoid arthritis (RA) and non-metastatic breast cancer).

These results are quite in line with previous research conducted in the general population, in individuals with chronic comorbidity; burnout and caregivers’ burden; cancer; pregnancy and parenthood and neurodegenerative diseases. A quantitative study conducted in rural Crete (the SPILI III project; n = 195, mean age 67.2 ± 15.2) found that higher SOC scores are related to lower Beck Depression Inventory Scale (BDI) scores, indicating a lower frequency of depression in individuals with high SOC (19). Similarly, the results for a Swedish cohort study of adolescent females with and without major depressive disorders (MDD) and anxiety disorders (n = 139, aged 14 to 18-year-old) led the authors to suggest that the SOC scale may be an inverse measure of depressive symptoms (43). In a Finnish study (n = 4642, aged 25 to 74-year-old), a high inverse correlation was obtained between SOC and depression (r = -0.62). Also, SOC was positively associated and depressive symptoms and anxiety were negatively associated with various health-related measures (44). A study investigating SOC in community-dwelling older Spanish adults (n = 1106, aged 60 years and older) revealed that higher SOC was significantly associated with higher functional status and has a moderate correlation with depression (r = -0.47, p < 0.001) (45). In a sample of 548 Japanese students (age range 18–37 years), SOC was negatively associated with mental health (46). The results of a clinical trial with depressive patients suggest that both rehabilitation and conventional depression treatment in a first episode of depression may enhance the SOC and rehabilitation itself enhances SOC more effectively in those with less severe depression (47). SOC has also been found to be a key psychological resource supporting the resilience of caregivers and their ability to cope with stressors. This in turn protects their psychological health. SOC remained relatively stable over a one-year period (48). It also mediates factors related to postpartum depression (PPD), so that intervention for enhancing SOC is recommended for women at risk of PPD (49). Definitely, the stronger the SOC, the fewer the symptoms of perceived depression (10).

Focusing on the research analyzed, it is found that chronic pain patients having a low level of SOC have more depressive symptoms, higher pain rates and a poorer level of functioning, less acceptance and less adaptation ability (23–29). Only one study failed to uncover an association between SOC and depression (31). These results are in line with the literature on SOC and chronic pain. High SOC is related to lower mean scores in pain intensity (50) and a weak SOC increases the likelihood of sustained pain and functional problems (51).

SOC is an important health-promoting resource that induces a positive perceived state of wellbeing (13). Strong SOC also allows people to identify, use, and ‘recycle’ available resources and therefore, minimize feelings of tension and stress (52). So, a strong SOC improves one’s ability to deal with health problems (12,53,54). Some findings provide initial evidence that SOC is a significant unique predictor of symptom change (55). Individuals with a weak SOC appear to be less optimistic and display less learned resourcefulness, self-efficacy, hardiness, locus of control, mastery, self-esteem, acceptance of disability, and social skills (13). SOC has a mediating effect on the influence of positive psychological constructs, life experiences, and social systems on our well-being (56).

Adopting a systematic salutogenic orientation to focus on available cognitive and emotional resources, strengthen them and create new ones could be very useful (12,51). The results encourage further clarification of the role and use of SOC in the rehabilitation context to minimize feelings of tension and improve function in everyday life (52). Also, making individuals more resilient is of imperative need during the current pandemic situation and further research of the SOC issues in relation to depression seems to be a high priority."

---

## [Decision Letter · Decision Letter 2]

19 Dec 2022

Association between sense of coherence and depression in patients with chronic pain: a systematic review and meta-analysis.

PONE-D-21-14267R2

Dear Dr. Oliván-Blázquez,

We’re pleased to inform you that your manuscript has been judged scientifically suitable for publication and will be formally accepted for publication once it meets all outstanding technical requirements.

Kind regards,

Thiago Machado Ardenghi

Academic Editor

PLOS ONE

Additional Editor Comments (optional):

Reviewers' comments:

Reviewer's Responses to Questions

**Comments to the Author**

1. If the authors have adequately addressed your comments raised in a previous round of review and you feel that this manuscript is now acceptable for publication, you may indicate that here to bypass the “Comments to the Author” section, enter your conflict of interest statement in the “Confidential to Editor” section, and submit your "Accept" recommendation.

Reviewer #3: All comments have been addressed

Reviewer #4: All comments have been addressed

2. Is the manuscript technically sound, and do the data support the conclusions?

Reviewer #3: Yes

Reviewer #4: Yes

3. Has the statistical analysis been performed appropriately and rigorously? 

Reviewer #3: Yes

Reviewer #4: Yes

4. Have the authors made all data underlying the findings in their manuscript fully available?

Reviewer #3: Yes

Reviewer #4: Yes

5. Is the manuscript presented in an intelligible fashion and written in standard English?

Reviewer #3: Yes

Reviewer #4: Yes

6. Review Comments to the Author

Reviewer #3: The authors have improved the manuscript, the quality of the manuscript has been greatly improved, and the current content has met the academic requirements of PLOS One.

Reviewer #4: This article falls within the scope and acceptance criteria of the journal. The aim is valid and relevant to the area. Methods are appropriate and reliable. Data is shown appropriately and the results support the conclusion.

7. PLOS authors have the option to publish the peer review history of their article (what does this mean?). If published, this will include your full peer review and any attached files.

Reviewer #3: No

Reviewer #4: **Yes: **Ângela Dalla Nora

---

## [Editor Report · Acceptance letter]

2 Jan 2023

PONE-D-21-14267R2 

Association between sense of coherence and depression in patients with chronic pain: a systematic review and meta-analysis 

Dear Dr. Oliván-Blázquez:

I'm pleased to inform you that your manuscript has been deemed suitable for publication in PLOS ONE. Congratulations! Your manuscript is now with our production department. 

Kind regards, 

on behalf of

Dr. Thiago Machado Ardenghi 

Academic Editor

PLOS ONE